# Slipknotted and unknotted monovalent cation-proton antiporters evolved from a common ancestor

**Vasilina Zayats**[1], **Agata P. Perlinska**[1], **Aleksandra I. Jarmolinska**[1,2],
**Borys Jastrzebski**[1], **Stanislaw Dunin-Horkawicz**[1], **Joanna I. Sulkowska**[1]*

**1** Centre of New Technologies, University of Warsaw, Warsaw, Poland, **2** Faculty of Mathematics, Informatics and Mechanics, University of Warsaw, Warsaw, Poland

* jsulkowska@cent.uw.edu.pl

## Abstract

While the slipknot topology in proteins has been known for over a decade, its evolutionary origin is still a mystery. We have identified a previously overlooked slipknot motif in a family of two-domain membrane transporters. Moreover, we found that these proteins are homologous to several families of unknotted membrane proteins. This allows us to directly investigate the evolution of the slipknot motif. Based on our comprehensive analysis of 17 distantly related protein families, we have found that slipknotted and unknotted proteins share a common structural motif. Furthermore, this motif is conserved on the sequential level as well. Our results suggest that, regardless of topology, the proteins we studied evolved from a common unknotted ancestor single domain protein. Our phylogenetic analysis suggests the presence of at least seven parallel evolutionary scenarios that led to the current diversity of proteins in question. The tools we have developed in the process can now be used to investigate the evolution of other repeated-domain proteins.

## Author summary

In proteins with the slipknot topology, the polypeptide chain forms a slipknot—a structure that is not necessarily manifest to a naked eye, but it can be detected using mathematical methods. Slipknots are conserved motifs often found at catalytic sites and are directly involved in molecular transport. Although the first proteins with slipknots were found in 2007, many questions remain unanswered, e.g. how these proteins appeared, or whether the slipknotted proteins evolved from unknotted ones or *vice versa*. Here we provide the first analysis of homologous slipknotted and unknotted transmembrane proteins in order to elucidate their evolutionary relationship. We show that two-domain slipknotted and unknotted membrane transporters share the same one-domain unknotted protein as an ancestor. The ancestor gene duplicated and underwent various diversification and fusion events during the evolution, which have led to the appearance of a large superfamily of secondary active transporters. The slipknot motif seems to have been created by chance after a fusion of two single domain genes. Therefore, we show here that the slipknotted

**Data Availability Statement:** All relevant data are within the manuscript and its Supporting information files.

**Funding:** JIS was supported by the EMBO (Installation Grant # 2057) and MNiSW (#0003/ID3/2016/64) and Polish National Science Centre (grant # UMO-2018/31/B/NZ1/04016). AIJ was supported by National Science Centre (grant #2018/29/N/NZ2/02897). SDH was supported by Polish National Science Centre (grant #2015/18/E/NZ1/00689) and FNP FIRST TEAM # 2018-2/14. The funders had no role in study design, data collection and analysis, decision to publish, or preparation of the manuscript.

**Competing interests:** The authors have declared that no competing interests exist.

transporter evolved from an unknotted one-domain protein and that there are at least seven different evolutionary scenarios that gave rise to this large superfamily of transporters.

## Introduction

Application of mathematical methods in the structural biology revealed that backbones of some proteins are entangled [1, 2]—they tie and form knots when pulled by their termini (Fig 1). These knots can be found both in globular and in membrane proteins [3]. Another type of an entanglement are slipknots, described by Todd Yeates, who originally detected them, as so: "*Slipknots occur when the path of some part of the chain forms a knot, which is then effectively undone when the terminus doubles back on itself, like a tied shoelace.*" [4].

The biological significance of a slipknot topology is not yet known, but it has been found previously in active sites of globular proteins [4, 5], and can be involved in transport mechanisms of transmembrane proteins [6]. Mechanical investigation of such proteins has shown that slipknot topology can give them a very high mechanical resistance [7], although such properties strongly depend on the slipknot geometry and the amino acid sequence motifs [8, 9]. The folding process of slipknotted proteins also remains unclear, and the question of whether its topology is a rate-limiting factor is unanswered [10, 11]. Theoretical investigations have shown that a slipknot could be formed either by flipping of a twisted loop (the blue loop, Fig 1) above a knotted core, or by pushing a slipknot loop (the orange loop, Fig 1) through a knotted core [10]. On the other hand, it was shown that the slipknot topology can appear as an intermediate step facilitating the entanglement in the case of the knotted proteins in the bulk [1, 12–18], in the confinement [19, 20], and in the case of a direct folding on the ribosome [21, 22]. The slipknot intermediate was also observed during unknotting [17, 23]. Finally, the slipknot topology has been observed as a step during the folding of other proteins with non-trivial topology such as lassos [24–27] and links [28, 29].

There is an increasing number of known proteins that form a slipknot or a knot in their native folded structure, however, their fraction is still very small [1]. Is it because they being are eliminated due to inefficient folding or structural complexity, and yet they make a sufficient contribution to protein structure that they are being preserved [30–32]? It was shown that both globular and transmembrane slipknotted proteins can provide an advantage in some extreme conditions [27], or increase the resistance to antibiotics [33, 34]. Still, the evolutionary origin of non-trivial topology in proteins remains unknown [35]. In this work we bring the field one step closer to answering these questions, by finding the first unknotted homologues to slipknotted membrane proteins.

Slipknotted transmembrane (TM) proteins were first discovered in 2007 [4], and since then many new structures with this topology have been discovered [6]. There are 16 different families with very different slipknot topology [36], but none of them have unknotted homologues. In consequence, the evolution of the slipknot could not be investigated. Therefore, we conducted a comprehensive analysis of all known protein structures from the KnotProt 2.0 database [36]. We identified proteins from sodium-dependent citrate symporter family (CitS, Fig 1B) as slipknotted, which remained unnoticed until now. A look at the fold of these proteins shows that they are made up of two structurally similar inverted domains, connected by a linker (S2 Fig). Other known membrane transporters with the slipknot topology are also composed of two inverted domains which, while very diverse in the sequence, are well conserved structurally. It was also shown that the majority of large transmembrane proteins are

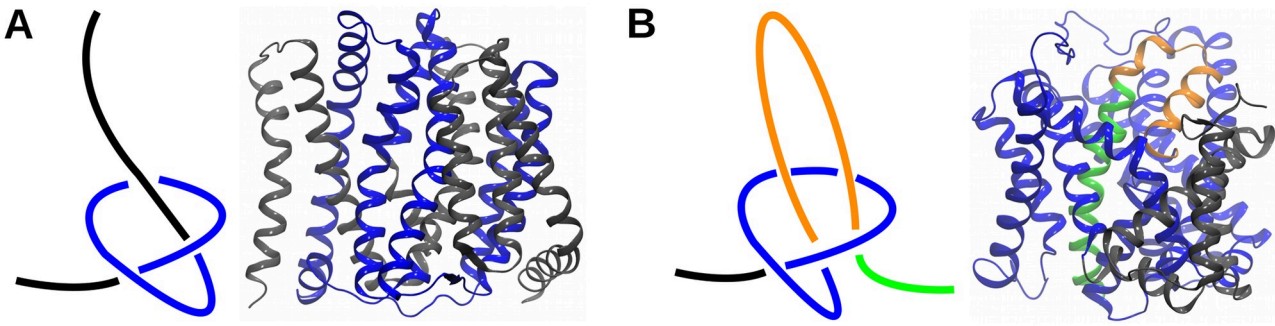

**Fig 1. Schematic representation of the knotted (A) and slipknotted (B) chains.** (A) The knotted chain (left) and protein backbone (PDBID: 4kpp) (right) with a knotted core marked with blue color. This knot is called open trefoil. (B) The slipknotted chain (left) and its representation in the sodium-dependent citrate symporter (KpCitS), a protein with slipknotted backbone identified in this paper (PDBID: 5xar). The knotted core is shown in blue, the slipknot loop in orange and the slipknot tail in green.

composed of duplicated domains positioned within membranes either parallel or anti-parallel to each other [37]. It was hypothesized that such proteins evolved from a one-domain ancestor gene *via* gene duplication and fusion [38, 39]. Thus, one could ask if the similarity between domains of slipknotted and unknotted proteins could be used to understand the evolution of the slipknot topology.

Based on the newly identified family of slipknotted proteins (sodium-dependent citrate symporter (CitS)), we aim to answer a long-standing question of how the transmembrane slipknotted structures appeared during evolution. Did they evolve from the unknotted proteins or vice versa? To test this possibility, we study the similarity between the domains of slipknotted and unknotted structures, and seek key elements that lead to the emergence of the slipknotted topology. Thus, we perform a thorough sequential and structural characterization of all homologous families. To study the characteristics of the domains, we utilize information from Pfam database [40], which recognizes proteins with related regions or domains as families enabling the research into evolutionary origin of these proteins. Based on a combination of known and new computational methods that we have developed, we analyze both sequence and structure of membrane transporters. This study provides evidence that several transmembrane protein families with different folds and topology types (slipknotted and unknotted) evolved from a common ancestor (schematically presented in Fig 2).

## Results and discussion

### New slipknotted family and the definition of the slipknot topology

Without going into much detail about entanglement detection methods, which have been extensively described in the literature [41–43], we introduce two terms important for the description of a slipknot: a knot core and a slipknot loop, as shown in Fig 1B. A slipknot can be imagined as knot of which one end doubles back through the loop leading to an ultimately unknotted structure. With that in mind, the knot core is the minimal part of the structure that would tie if pulled by its ends, and the slipknot loop is the "doubling back" part of the chain, as shown in Figs 1 and 2. In general, slipknots in proteins can be very complicated, they can be formed based on different types of knots along a single chain [6].

In the case of newly identified proteins with PDBID 5a1s (SeCitS) and 5xar (KpCitS) discussed in the next section, we found that amino acids between 166–411/115-401 (respectively) form a knot core (the blue loop) and amino acids between 412–415/402-422 (respectively) form a slipknot loop (the orange loop), as shown in Figs 1 and 3. The knot core forms the knot

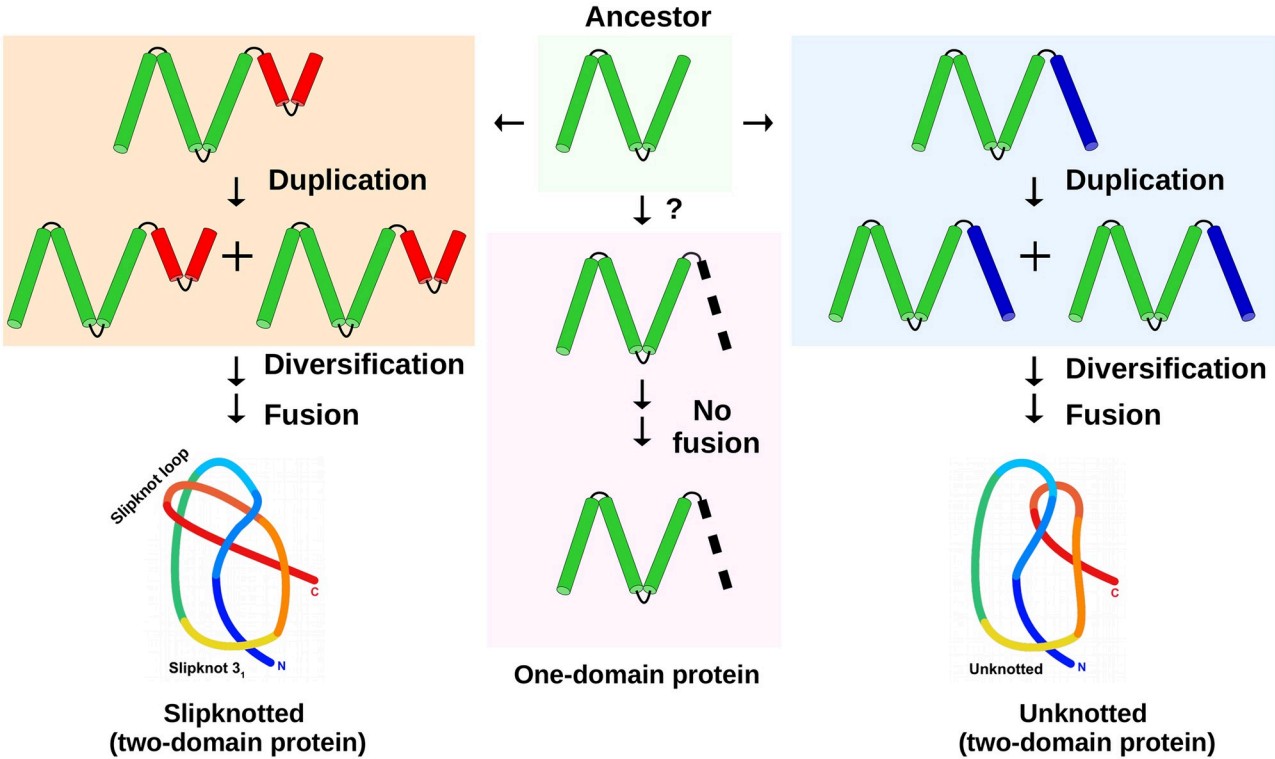

**Fig 2. Conserved helical region (core) found in the monovalent cation-proton antiporter superfamily.** Conservation of this region suggests that three different fold types, including one possessing a non-trivial topology (a slipknot), evolved from a common, single-domain ancestor. The putative ancestor is shown in light green box in the middle. Three arrows from the ancestor navigate to three proteins with different folds: 1) left—two-domain slipknotted protein; 2) middle—one-domain unknotted protein; 3) right—two-domain unknotted protein. On the bottom left of the figure is shown a schematic diagram of the entangled region of the slipknotted protein colored from blue (N-terminal) to red (C-terminal). On the bottom right there is a similar schematic diagram that shows the topology of the unknotted protein backbone.

called a trefoil which is denoted $3_1$ (it possesses three crossings when projected on the plane). Altogether, this protein forms a $3_1$ slipknot, however, for simplicity in this paper we will just call it a slipknot.

## Slipknotted family of sodium-dependent citrate symporters (CitS) belongs to the monovalent cation-proton antiporter superfamily

By using the KnotProt 2.0 database we spotted a slipknot topology in two proteins (PDBID: 5xar and 5a1s) from a family never before identified as entangled. In both proteins (sharing 92% similarity) the slipknot is formed by a single trefoil knot (the blue loop in the Fig 3) where the C-terminus doubles back on itself forming the orange loop, as shown in the Fig 3. These proteins are found in the cell membranes acting as transporters (sodium-dependent citrate symporter—CitS). They belong to the 2-hydroxycarboxylate transporter family (2HCT) as recognized by Pfam database (PF03390). A careful examination of the structures shows that they are composed of two structurally similar domains (S2 Fig).

In order to find all the proteins related to this slipknotted family, we used HMMER web server [44]. Altogether, we found 17 protein families, which are all parts of the same monovalent cation-proton antiporter superfamily according to the OPM database [45] (Table 1). Four of these families are represented by proteins with known 3D structures, and thus topology. Three unknotted families are Sodium/hydrogen exchanger family (represented by PDBID:

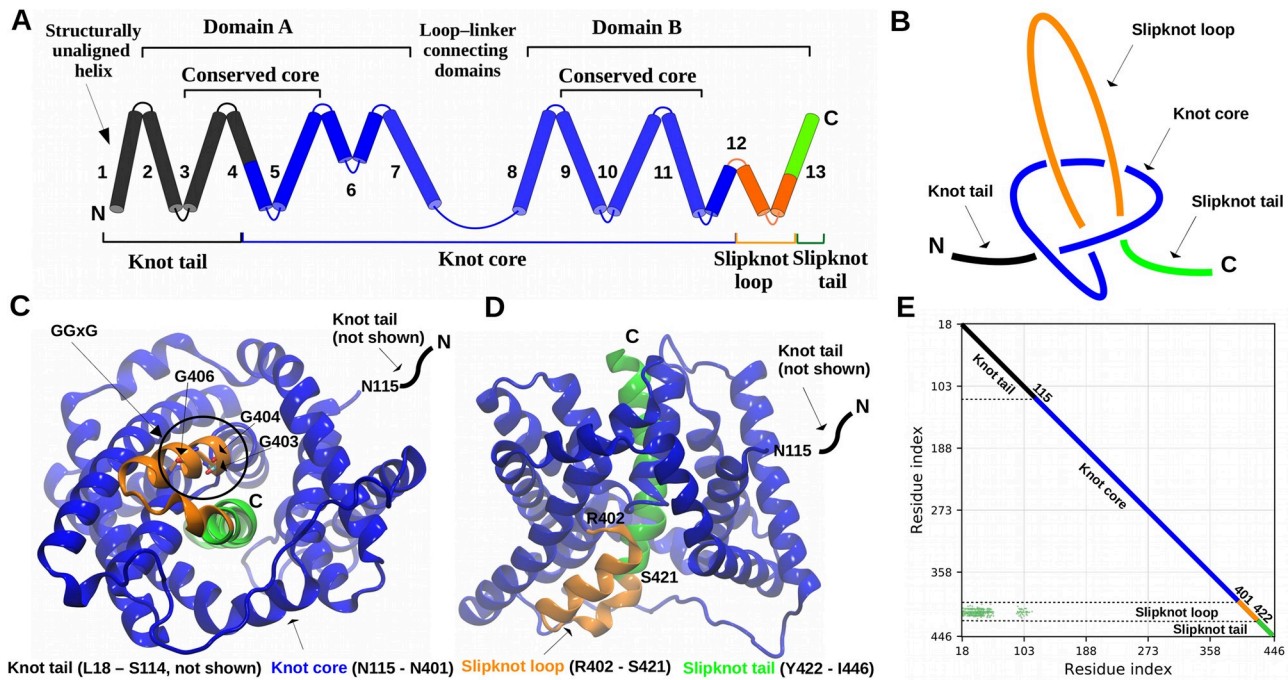

**Fig 3. Structure of the slipknotted transporter KpCitS.** (A) Location of the knotted core (N115-N401) starts at TM4 of domain A, loop-linker and TM8-TM11 and half of TM12 hairpin-like helix of domain B. Slipknot loop (R402-S421) is formed by half of TM12 hairpin-like helix and part of TM13. The rest of TM13 (residues Y422-I446) is slipknot tail. (B) Schematic representation of the slipknot topology. (C, D) Structure of the knotted core and slipknot's loop and tail based on PDBID 5xar. (E) Knot fingerprint calculated based on the KpCitS structure (PDBID: 5xar).

**Table 1. Known protein families from monovalent cation-proton antiporter superfamily investigated toward identification of possible evolution of the slipkotted topology.** IDs of families and clans are from Pfam database. ND—not determined.

| Family ID | Clan ID | Name | Topology | Number of domains |
|---|---|---|---|---|
| PF03390 | - | 2-hydroxycarboxylate transporter family | slipknotted | 2 |
| PF00999 | CL0064 | Sodium/hydrogen exchanger family | unknotted | 2 |
| PF06965 | CL0064 | Sodium–hydrogen antiporter 1 | unknotted | 2 |
| PF01758 | CL0064 | Sodium Bile acid symporter family | unknotted | 2 |
| PF03547 | CL0064 | Membrane transport protein | ND | 2 |
| PF03601 | CL0064 | Conserved hypothetical protein 698 | ND | 2 |
| PF03812 | CL0064 | 2-keto-3-deoxygluconate permease | ND | 2 |
| PF03977 | CL0064 | Na+-transporting oxaloacetate decarboxylase beta subunit | ND | 2 |
| PF03956 | CL0064 | Lysine exporter LysO | ND | 2 |
| PF05684 | CL0064 | Protein of unknown function (DUF819) | ND | 2 |
| PF05982 | CL0064 | Na+-dependent bicarbonate transporter superfamily | ND | 2 |
| PF03616 | CL0064 | Sodium/glutamate symporter | ND | 2 |
| PF13593 | CL0064 | SBF-like CPA transporter family (DUF4137) | ND | 2 |
| PF06826 | CL0064 | Predicted Permease Membrane Region | ND | 2 |
| PF05145 | CL0142 | Transition state regulatory protein AbrB | ND | 2 |
| PF03788 | - | LrgA family | ND | 1 |
| PF04172 | - | LrgB-like family | ND | 1 |

4bwz), Sodium/hydrogen antiporter 1 (represented by PDBID: 1zcd) and Sodium Bile acid symporter family (represented by PDBID: 3zuy). The fourth family is represented by the newly identified slipknotted proteins with PDBID 5a1s and 5xar. According to our knowledge this is the first clan where proteins with different topology were identified. This finding allows tracing and investigation of possible evolutionary steps to forming the slipknotted family. Proteins from the remaining families have undetermined topology since no structure was experimentally resolved.

## Structural comparison shows a common motif in both slipknotted and unknotted transporters

In order to characterize the relationship between slipknotted and unknotted transmembrane proteins we have conducted their structural analysis. Both slipknotted and unknotted transporters are made up of two inverted structurally similar domains. Therefore, for the analysis, we extracted the domains (A and B) and compared them. As the representative structures, we have chosen unknotted structure of sodium-proton antiporter NhaA (PDBID: 4bwz, from PF00999 family) and slipknotted structure of 2-hydroxycarboxylate transporter SeCitS (PDBID: 5a1s). In both proteins the domains are created by 6 transmembrane helices (TM)—TM2-TM7 and TM8-TM13 in the slipknotted structure (domain A and B, respectively; Fig 4A) and TM1-TM6 and TM8-TM13 in the unknotted protein (domain A and B, respectively; Fig 4B).

First, we analyzed the level of structural similarity of the domains from the same structure. Domains from the slipknotted CitS slightly differ in length—domain A is 189 amino acid long (from T49 to Y237) and domain B is shorter by 5 residues (H265 to E448). The domains of the unknotted protein are slightly shorter than those of its slipknotted counterparts since both of them are 172 amino acid long (domain A—G3-G174 and domain B—P215-E386). Superimposition of the domains from the same protein shows that the structural similarity between domain A and B is high for both proteins (RMSD of 2.2 Å over 74 aligned residues for slipknotted and 68 for unknotted protein). However, comparison of their sequences shows that they are substantially different. The domains of slipknotted SeCitS are identical only in 15.2% (similar in 27.2%). The results for the unknotted protein show a comparable level of identity (14.5%) and similarity (27.3%). Therefore, for both proteins, regardless of their topology, we observe high structural similarity of the domains with low sequence conservation.

Next, we analyzed whether the structural similarity of the domains is present also between the proteins. We found that in general, the domain of the slipknotted protein can be easily superimposed on the domain of the unknotted one (Fig 4D). Although spatial structures of the domains differ, there is a TM-helical region (core) which is structurally similar in both domains (RMSD of approx. 2Å over 80 aligned residues; Fig 4D). This region consists of 3 transmembrane helices. In the slipknotted structure it is located between TM3 and TM5 for domain A (G80-C165) and between TM9 and TM11 (H299-G385) for domain B. For unknotted protein the core is placed between TM2-TM4 (P30-G110) for domain A and TM9-TM11 (P237-T315) for domain B.

The difference between the domains of slipknotted and unknotted proteins is manifested in the conformation of TM6 from the slipknotted protein and the corresponding TM5 from the unknotted one (Fig 4). TM5 from the unknotted structure goes straight through the membrane, while TM6 from slipknotted protein bends in the middle because of its GGxG motif and creates a hairpin-like structure (Figs 3C and 4F). Therefore, it changes the direction of the polypeptide chain. Most importantly, the motif is conserved in the slipknotted family in both domains and is absent in unknotted proteins (S1 Fig). Consequently, the following C-terminal TM helices end up at opposite sides of the membrane.

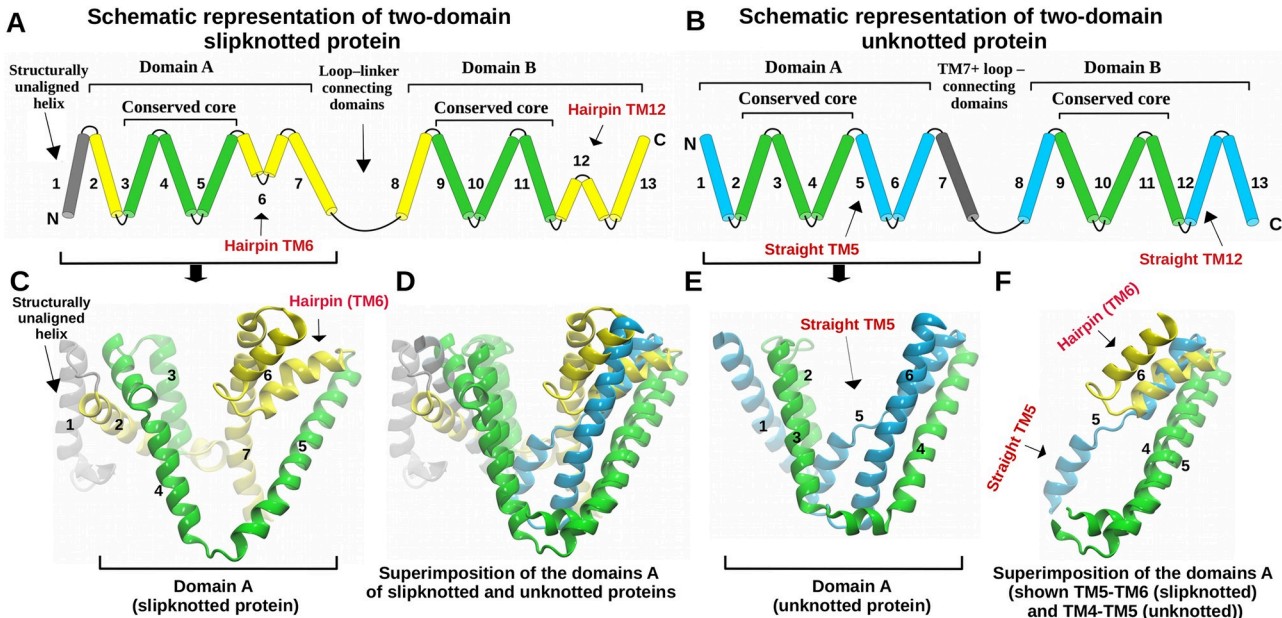

**Fig 4. Identified structural differences between the slipknotted and unknotted proteins.** (A) Schematic representation of the full length two-domain slipknotted protein. Two domains are connected by the long loop. The conserved core region is colored in green. TM helices are enumereted as in the structure (PDBID: 5a1s). (B) Schematic representation of the unknotted protein. Similarly, unknotted protein is composed of two domains. Conserved core is colored in green. TM helices are enumereted as in the structure (PDBID: 4bwz). (C) Structure of a single domain of the slipknotted protein (PDBID: 5a1s). (D) Superposition of the domains A of the slipknotted and unknotted proteins. (E) Structure of a single domain of the unknotted protein (PDBID: 4bwz). (F) Superposition of the fragments of domains A of the slipknotted and unknotted proteins.

The difference in the topology between slipknotted and unknotted structures is most probably caused by the aforementioned transmembrane helices (TM5 and TM6), as well as by the linker between the domains (Fig 4 and S2 Fig). In the structure of the unknotted protein, there is an additional TM7 helix and a short loop which connects the two domains. On the other hand, in the slipknotted structure, domains are connected by a long (30 amino acids) cytoplasmic loop, which makes a turn around the structure (S2 Fig). This long flexible loop most likely plays an important role in forming the slipknot topology, as it wraps like a lasso around the structure to form the knot.

## Comparison of sequence profiles reveals complex connections within the superfamily

In order to trace the evolution of the slipknotted transporter and related proteins, we broadened our investigation and performed sequence analysis based on all protein families. We took all the protein sequences from 17 families and for each domain we calculated multiple sequence alignment (MSA) and a profile (Fig 5A, S3 and S4 Figs and S2 File). Next, based on the profile-profile comparison of separated domains of each family, we traced the evolutionary events that resulted in the present day diversity (see Fig 5B and S3 Fig). Domains within the slipknotted family showed weak similarity to each other. Domain B is most closely related to domain A from PF03601, to domain B from PF05684 and to LrgA (PF03788), a one-domain family. In contrast, a sequence of domain A from the slipknotted protein differs significantly from all other domains in our dataset, including its sister domain. The same lack of similarity between two domains belonging to the same family is also seen for several other families (PF01758, PF03547 and PF013593; Fig 5A and S3 Fig). Interestingly, these families have high

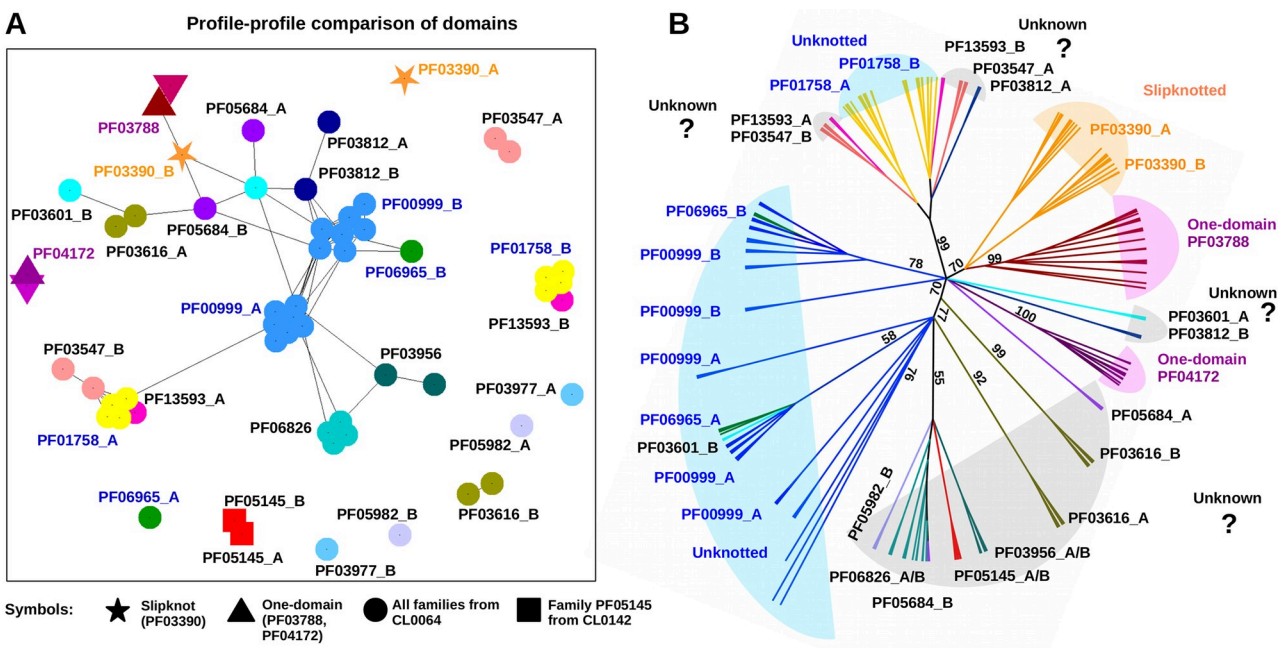

**Fig 5. Sequence and phylogenetic analysis of the domains suggests a common evolutionary origin.** Panel A shows clustering of domain profiles. Each family shown in a different color, connections indicate similarity found with e-value below $1e^{-5}$. Two domains of the slipknotted protein are shown as orange stars. All families from CL0062 are shown as circles colored according to the family. The family IDs with known unknotted topology are highlighted in blue font. The one-domain proteins (PF03788 and PF04172) are shown as triangles and family ID are colored in magenta font. Two domains of PF05145 (CL0142) are shown as red squares. Panel B shows the phylogenetic tree of the domains. Color-coding of tree branches is the same as for domains clustering on the panel A. Additionally, colored background areas serve to group the tree branches according to some properties: 1) families with known unknotted topology are highlighted with blue; 2) the slipknotted family is highlighted in orange; 3) the families with unknown topology are highlighted in light gray; 4) the families of one-domain proteins are highlighted with magenta.

sequence similarity based on the full-sequence clustering (S10 Fig). When divided, domains A and B are clustered separately, which indicates that the two regions diverged over time (Fig 5A and S3 Fig, yellow, magenta and pink circles, respectively). Moreover, domains from PF03547 are clustered with their counterparts from other families (domain A in a group with domains B and vice versa; Fig 5A and S3 Fig). This indicates that proteins belonging to this family are products of a reverse-order fusion.

It is worth noting that the lack of sequence similarity between domains does not occur in all protein families—the largest unknotted family (PF00999) has some of the domains A and B well connected in sequence based clustering, thus significantly similar (in sequence) to domains from other families and to themselves. Additionally, we also found outlaying protein families, that show significant similarity only between their own domains (PF06826, PF03601, and PF05145; Fig 5A and S3 Fig).

## Multiple sequence alignment shows a highly conserved core region

In order to directly compare the sequences of all the domains, we created a tool that generates a Multiple Profile Alignment (a multiple sequence alignment of the profiles). Only profile-profile alignments with e-value lower than $10^{-3}$ were used in creating it. Because of that, domain A from PF05982, as well as both domains from PF03977, were removed from the analysis, as they showed no statistically significant similarity to any other domain.

Multiple alignment (S4 Fig) shows that all of the remaining domains align well. The best conserved region of the alignment corresponds precisely to the transmembrane helical core we

discussed above, found in all 3D structures ([Fig 4]). It is feasible that the core serves as the functional unit and dates back to the common ancestor of all the analyzed proteins. Moreover, we found that N- and C-termini of the domains are the most diverse parts of the sequences. The N-terminal part often differs in the number of transmembrane helices between the families—from no helices in the structure from PF01758 family (PDBID: 1zuy), to one (PDBID: 4bwz) and two (PDBID: 4czb) in structures from PF00999. C-terminal part may be the region of subfunctionalization since it contains the helices located at the internal pore of transporters.

## Phylogeny shows seven distinct evolutionary paths resulting in internally repeated proteins

[Fig 5B] presents a diagram showing the evolution of the repeated regions found in membrane transporters (four versions of the tree are available in Supplementary Materials, Figs 5–8). In principle, all the families from our analysis could be represented in the same organism (our taxonomical analysis has shown up to 13 found in the same organism family; see [S9 Fig]).

In most cases, an ancestral gene duplicated and later, in the second event, the two genes fused to encode the two-domain protein. Between the events, the genes individually evolved, therefore we observe substantial differences between their sequences. However, another possible path that we observed is a single internal duplication event or a fast fusion. In this case, the duplication occurred within the gene, or the duplication and fusion were nearly simultaneous, thus leading to a protein with two domains with highly similar sequences.

Overall, our data shows seven evolutionary scenarios within the studied group of transporters (shown in Figs [5] and [6]):

1. No fusion—diversification of the ancestor gene leading to one-domain protein. Based on the profile analysis ([Fig 5A]) we found that one-domain families LrgA (PF03788) and LrgB (PF04172) are distantly related to two-domain slipknotted and unknotted proteins. Therefore, these families should share a common ancestor. The single-domain LrgA and LrgB families did not fuse, but interestingly, a LrgA-LrgB fusion protein was found in plants.

2. Fast fusion—gene duplication followed by an instant fusion (in families PF05145, PF06826 and PF03956). In these families domains A and B have very similar sequences. Domains A and B of the families PF05145 (red squares), PF06826 (cyan circles) and PF03956 (dark green circles) are located next to each other on the profile comparison figure ([Fig 5A] and [S3 Fig]). This indicates that domains duplicated and fused very soon after the duplication, therefore they are still very similar, since they did not have time to diverge.

3. Reverse-order fusion—gene duplication and fusion in a reverse order. Unknotted families PF03547 and PF01758 are closely related based on full sequence similarity ([S10 Fig]). However, when sequences are divided into two domains, another picture emerges—domain A of PF03547 is more similar to domain B of PF01758 than to its sister domain B and also to domain A of PF01758 ([Fig 5A] and [S3 Fig]). It means that after gene duplication domains A and B fused in different order than in other families (not AB but BA).

4. Long diversification—gene duplication followed by an extended period of diversification (subfunctionalization). In slipknotted family PF03390 and unknotted PF01758, most of PF00999 and PF03812, the two domains are not similar in sequence. This means that after gene duplication there was a long period of diversification before the domains A and B fused.

5. Duplication of already fused protein (PF13593 speciated from PF01758). Full sequences of families PF13593 and PF01758 are very similar, partly identical ([S10 Fig]). The same is true

**Evolutionary scenarios detected based on the sequence similarity**

**Fig 6. Seven evolutionary scenarios found within the transporters.** 1) No fusion—diversification of the ancestor gene leading to one-domain protein. 2) Long diversification—gene duplication followed by an extended period of diversification which lead to two domains with low sequence similarity (shown with red and blue). 3) Duplication of already fused protein (PF13593 speciated from PF01758, PF06965 from PF00999). 4) Reverse-order fusion—gene duplication and fusion in a reverse order. 5) Fusion of the domains from different lineages (shown with blue and green). 6) Fast fusion—gene duplication followed by an instant fusion (in families PF05145, PF06826 and PF03956). 7) Fusion of unrelated lineages (PF05982).

for the separate domains—domain A of PF13593 is most similar to domain A of PF01758, domain B of PF13593 is most similar to domain B of PF01758 based on profile-profile comparison (Fig 5A and S3 Fig). In this case, a full two-domain protein got duplicated and further evolved into two families PF01758 and PF13593. Therefore, these families share a common recent two-domain ancestor.

6. Fusion of the domains from different lineages formed (PF03601, PF05684, PF03616, PF06965). Here two domains of one protein do not share significant similarity to each other but, instead, are more similar to the domains from another family. For example, in family PF03601 domain A is more sequentially similar to domains from families PF05684, PF03812 and PF03390 than to its sister domain B.

7. Fusion of unrelated lineages formed (PF05982). In this family domain B is not similar to any other domain from our dataset. It is not clear whether the second domain originated from the same common ancestor or not. One possible scenario is that domain B strongly diverged and sequence similarity is not detectable anymore. Another possible scenario is that domain B evolved from a different unrelated clan. It would be possible to determine which scenario is correct when the structure from this family is available.

In order to understand which families appeared earlier in the course of evolution, we projected the superfamily tree on the Tree of Life (S9 Fig). The results show that PF00999 and PF01758 families are the most widely distributed across the species. They are present in all three domains of life (Bacteria, Archaea and Eukaryota). Therefore, these two families are perhaps the oldest among the superfamily. On the other hand, families PF05145 from the clan CL0142 and slipknotted PF03390 are not found in Archaea, which indicates that these families evolved later. This is also supported by the fact that PF05145 and PF03390 represent two different Pfam clans, outside of the main CL0064. Therefore, our data suggest that the family with slipknot topology (PF03390) emerged later in evolution. Family PF03812 has only been found in Bacteria (however this might be due to lack of data). The fusion of LrgA and LrgB must have happened later in evolution since it is found only in plants.

Interestingly, according to identified phylogenetic tree and sequence profiles, one-domain LrgA is most closely related to domain B of the slipknotted family. Due to the fact that LrgA family is widely present on the tree of life while the slipknotted one is not, LrgA might evolved earlier than domain B. This suggests that the family with slipknot and one-domain LrgA share a common recent ancestor (Fig 5B and S5–S8 Figs).

Our profile analysis shows that there are also two-domain families that are significantly similar to the family of slipknotted proteins—PF03616, PF03601 and PF05684. Due to the lack of experimental data and resolved structures, the topology of these families' proteins is not known. Based on our results we suggest that proteins homologous to slipknotted PF03390 could possess non-trivial topology as well. The same conclusion was derived earlier based on experimental studies and sequence similarity for one of the families (PF03616) [46, 47].

## Potential function of slipknot topology

It has been shown that the position of the active site very often overlaps with the location of the knot [6]. The universal role of such topological constraints is not known, however, in the case of some families its direct influence on biological function has been observed [48–50].

There is no recent and up to date review article describing slipknotted proteins. Our search through the slipknotted proteins deposited in the KnotProt database shows that slipknot topology, and more precisely the slipknot loop, is located directly in the active site of many globular

and transmembrane proteins [36]. Even though, the role of the slipknot loop is unknown for the family studied here, we can say that it is directly involved in the transport mechanism (Fig 3). From the available structures, captured in different conformations (inward and outward open), it can be seen that the movement of the hairpins is connected with the transport of molecules across the membrane [51]. Hairpin formed by TM12 is also a slipknot loop and contains the conserved GGxG motif (Figs 3 and 4). Both hairpins TM6 and TM12 are involved in the coordination of sodium ions and substrates. Thus the slipknot constraints could be responsible for strapping together the transmembrane helices to form a flexible but stable channel similarly as was suggested in [6]. Therefore, slipknot topology is likely important for the transport mechanism of the PF03390 family of transporters.

## Possible mechanism of slipknotted membrane protein folding

Proteins with complex topology such as slipknot pose a challenge for folding theories [6, 52]. In general, there are at least three ways to form a simple slipknot topology. First, it could be formed randomly during protein folding and then be stabilized and moved to the native position. In the case of knotted proteins, it was shown that such a scenario is very unlikely. However, theoretically one could imagine that such random slipknot could be stabilized when packed inside a membrane.

The second mechanism is based on the formation of the knot first and then one terminal would have to be threaded back the knotted loop to form the slipknot loop. This mechanism involves crossing the topological barrier twice, thus it is very unlikely for both globular and membrane proteins.

Third, the slipknot topology can be created by the formation of a twisted loop through which slipknot loop passes, thus bypassing the knotted structure stage altogether. Indeed such flips of the loop over the protein's core to form a slipknot motif were observed in the case of globular slipknotted proteins. Moreover such flipping (called also a mousetrap) was recognized as the key mechanism in knotting a protein with a Stevedore knot ($6_1$ type) [53], the Gordian knot ($5_2$ type) [54–56], as well as one of possible ways to form trefoil knot ($3_1$ type) [10, 17, 31, 57, 58].

Comparing the described mechanism with our slipknotted membrane protein one could see some similarity. The loop that flips over a portion of the protein in other slipknotted proteins such as in a thymidine kinase [10] can be equated with the linker found in slipknotted protein family studied here (CitS; Fig 3). Even though we are not sure when, during the folding process, such a flipping could happen, one possibility is to first form the twisted core of the protein and then push the slipknot loop through it during insertion into the membrane. The linker is missing in unknotted families, suggesting that these proteins fold in a different way.

As the question of slipknot folding remains open, other pathways are being proposed [11]. However, this pathway is the most complex one and further investigations are necessary to understand how it could be applied for transmembrane proteins.

## Conclusions

We identified the family of transmembrane proteins (2HCT transporters, PF03390) as possessing non-trivial slipknotted topology, which was not reported before. Moreover, we found that this family is related to several other families that contain unknotted proteins. This unique case allowed us to conduct a qualitative and quantitative investigation of the evolutionary relationship between slipknotted and unknotted proteins from the monovalent cation-proton antiporters superfamily and to understand how the slipknot topology appeared. Based on our sequential and structural analysis of the proteins' repeated domains, we established that both

slipknotted and unknotted proteins evolved from a common one-domain ancestor (Fig 2). The distant homology is further supported by the existence of a common core region of three transmembrane helices found in all known structures from the studied families (regardless of the topology; Fig 4).

Intriguingly, we found that domains of three two-domain families (PF03616, PF03601 and PF05684) are similar in sequence to the slipknotted domain (Fig 5A and S3 Fig). Therefore, these proteins might possess non-trivial topology as well.

Our analysis shows that the current diversity of membrane transporters was achieved through several evolutionary scenarios that allowed for diversification from a common, one-domain ancestor. Therefore, other transmembrane proteins with slipknot and knot topology could have followed similar paths. Our analysis indicates that the evolution of two-domain slipknotted family 2HCT started with gene duplication, and after a long diversification period the two genes merged. It was only the fusion of two genes (coding unknotted proteins) that made this slipknot protein possible.

It has been shown that the knotted proteins, both membrane and globular, can also consist of inverted repeats [3]. It is worth noting that the first artificially knotted protein was constructed based on two inverted repeats, thus our finding, supported by newly developed bioinformatics methods, could be used to identify and design artificially knotted proteins [59].

We also briefly discuss possible folding of slipknotted membrane proteins, suggesting that identified flexible linker could facilitate transition from unknotted to slipknotted topology. We also point out a potential role of slipknot topology in discussed family.

To our knowledge this is the first research devoted to the homologous slipknotted and unknotted transmembrane proteins. The tools we have developed can now be used to investigate the evolution of other proteins.

## Methods

### Data set

We used profile search HHMER (Jackhmmer [44]) to find homologues of the protein families from Monovalent cation-proton antiporter superfamily. Our final data set comprised of 17 Pfam [60] families: PF03390 (slipknotted, two-domain), 13 (two-domain) families from the CPA_AT clan (CL0064: unknotted PF00999, PF06965 and PF01758; families of unknown structure/topology: PF03547, PF03601, PF03812, PF03977, PF03956, PF05684, PF05982, PF03616, PF13593, PF06826), PF05145 from CL0142 (structure unknown), and two families with only one domain—LrgA (PF03788) and LrgB (PF04172) (structure unknown), and sequences of the fusion LrgA/LrgB.

### Sequence analysis

All sequences in the data set were filtered at 90% similarity, and only sequences (for two-domain proteins) with lengths deviating by at most half of their length from the average length for a given group were kept. This left us with 28 717 sequences. The families differ significantly in the number of sequences, as can be seen in the S10 Fig. For example, the largest family PF00999 counts 11708 sequences (in the final dataset), slipknotted family PF03390 counts 295 sequences (in the final dataset). After filtering, the sequences were clustered by similarity (using CLANS [61]) and aligned within each cluster (using PROMALS3D [62]). Each resulting alignment was then separated into domains based on the alignment to one of the known protein structures (representatives from PF00999, PF06965, PF01758 and PF03390 families). Each of the single-domain-alignments was used to create a sequence profile (output files from

clustering of domain profiles can be found in the S3 File; due to the file size, the output showing full-length sequences can be obtained upon request).

Sequence profiles for each domain—that is a combination of protein family, domain within the sequence (N- or C-terminal if applicable) and CLANS cluster within that family, were used to create a Multiple Profile Alignment (MPA). As, to our knowledge, no software with this capability is available, we have created an in-house script [63] (source code at https://github.com/ilbsm/HHsearch-results-aligner) based on the principle of the maximum weight trace. As an input, we used pairwise profile-profile alignments generated by HHsearch, with the final MPA optimizing the agreement between them. Additionally, this approach simplifies the resulting alignment by removing sequence fragments that cannot be gainfully aligned to anything (thus expand the alignment without providing any additional data). The MPA was then transformed into a multiple sequence alignment by replacing the profile states with corresponding residues of a representative sequence from each domain (selected based on its agreement with the profile).

### Phylogenetic tree construction

Finally, the multiple sequence alignment of the separated (i.e. treated as unrelated) domains was used to create a phylogenetic tree (using MrBayes; Fig 5B and S4 File) [64], with LG model and a characteristics matrix coding additional information from profile clustering results (S11 Fig) and N- and C-terminal regions similarity—for more details see Supplementary Materials (S5–S8 Figs and S1 File).

## Supporting information

**S1 Fig. Sequence logos of both slipknotted and unknotted protein families.** Sequence logo of slipknot family PF03390 showing that multiple glycines are highly conserved across the whole family. The logos were generated from families multiple sequence alignments (available in Pfam) with WebLogo3.
(TIFF)

**S2 Fig. The linkers connecting two domains in slipknotted and unknotted structures.** Figure shows that slipknotted and unknotted proteins are composed of two inverted domains which are connected by the linker. Panel A shows slipknotted structure (PDBID: 5a1s). From left to right: domain A, linker and domain B are shown. Similarly, panel B-C shows the linkers between the domains in unknotted structures.
(TIFF)

**S3 Fig. Profile-profile comparison of the domains.** Comparison of domains profiles, shown at cut-off 1e-5. Every domain profile is shown as one point with different shapes: star, circle, triangle, square. The connections between domains are shown as straight lines. The connection indicates that profile-profile alignment of these domains has significance value 1e-5 or less. Every family is colored in unique color, same as in the main Fig 5A. Two domains of the slipknotted family are shown as orange stars. All families from CL0062 are shown as circles colored according to the family. The families IDs with known unknotted topology are highlighted in blue font. The one-domain proteins (PF03788 and PF04172) are shown as triangles and families IDs are colored in magenta font. Two domains of PF05145 (CL0142) are shown as red squares. Families (PF00999 (blue), PF01758 (yellow), PF3547 (pink), PF03616 (olive), PF06826 (dark cyan) were divided into several subgroups based on full sequence clustering (S10 Fig), therefore there are more than two domains in these families. Domain A and B of PF00999 have separated into two clear clusters.
(TIFF)

**S4 Fig. Multiple sequence alignment of the domains profiles.** (A) Multiple alignment of domains profiles revealed the conserved core region. Y-axis lists all families which pass through the alignment threshold 1e-3. Every line in the alignment represents family profile. On the X-axis sequence length colored from blue to red. White spaces in the alignment are present when no significant similarity was found between the profiles. The borders of the domain and conserved 3 TM helical core are indicated below the plot. The turn in TM12 forming the slipknot loop is highlighted by black rectangle and next to it the multiple sequence alignment with conserved GGxG region is shown. (B) Schematic representation of individual domains of slipknotted and unknotted proteins.
(TIFF)

**S5 Fig. Bayesian phylogenetic tree 2.** The tree was generated from multiple sequence alignment of the domains using the characteristics matrix multiplied 10 times. The characteristics matrix (S11 Fig) was generated based on profile-profile connections (S3 Fig). For the tree calculation three representative sequences of each family were used. The tree shows several main branches: 1) Both domains of the slipknotted family PF03390 and one-domain family PF03788 are located on the same branch; 2) Another separated branch joins closely related families PF01758, PF013593 and PF3547; 3) Domains A and B of the unknotted family PF00999 were separated into two branches. Domains B of PF00999 are grouped together with the domain B of unknotted family PF06965 and with domain A of PF05684 (unknown topology). Domain A and B of families PF03956, PF06826, PF05145, domain B of PF05982 are located in one branch.
(TIFF)

**S6 Fig. Bayesian phylogenetic tree 3.** The tree was generated from multiple sequence alignment of the domains using the characteristics matrix multiplied 5 times. The characteristics matrix was generated based on profile-profile connections (Fig 5A and S3 Fig). The characteristics matrix is shown on S11 Fig. For the tree calculation 10 representative sequences of slipknotted PF03390 and one-domain families (PF03788, PF04172) and three representative sequences of remaining families were used. The tree shows three main branches: 1) domains A and B of slipknotted family PF03390 and one-domain family PF03788 are located together on one branch; 2) Another separated branch joins closely related families (domains A and B) PF01758, PF013593 and PF3547; 3) Domains A and B of the unknotted family PF00999 are located on different branches. Domain A of another unknotted family PF06965 is located together with domains A of PF00999 and domain B of PF06965 is placed together with domain B of PF00999.
(TIFF)

**S7 Fig. Bayesian phylogenetic tree 4.** The tree was generated from multiple sequence alignment of the extended conserved core (S4 Fig) which includes the conserved 3-TM helical core + 1 next TM helix (hairpin in slipknotted family). The characteristics matrix based on the profile-profile connections was used as in trees S5 and S6 Figs. Additionally, removed N- and C-terminal regions of the domains were introduced into tree calculation as N/C matrices. Characteristics matrix and N/C matrices were multiplied 10 times. For the tree calculation 10 representative sequences of slipknotted PF03390 and one-domain families (PF03788, PF04172) and three representative sequences of remaining families were used. The tree shows three main branches: 1) slipknotted family PF03390 is placed together with both one-domain families PF03788 and PF04172; 2) Also, as previously, a separated branch joins closely related families PF01758, PF013593 and PF3547; 3) Domains A and B of the unknotted family PF00999 are separated on the tree. However, evolution of other families is not resolved in this tree. PF04172

is placed on the same branch with PF03788 and according to profile analysis these families are distantly related. Also, PF03812(B) and PF03601(A) are together with PF00999(B) which is also in agreement with profile analysis.
(TIFF)

**S8 Fig. Parsimony phylogenetic tree.** The parsimony tree was generated from the extended core region (conserved core + 1 next TM helix (hairpin in slipknotted family)) same as in the Bayesian phylogenetic tree (S7 Fig). Eight families were used for this analysis: PF00999 (unknotted), PF01758 (unknotted), PF03547 (unknown topology), PF13593 (unknown topology), PF03390 (slipknotted), PF03788 (one-domain), PF04172 (one-domain) and two-domain family PF05145 from the clan CL0142. Five representative sequences of all eight families were used for tree construction. The tree shows four main branches with duplication events producing families: 1) slipknotted family PF03390 and one-domain family PF03788; 2) two-domain closely related families PF01758, PF013593 and PF3547; 3) unknotted PF00999 with two sub-branches separating domains A and B; 4) two-domain family PF05145 from another clan (CL0142) was placed on the separate branch. In this tree one-domain family PF04172 is placed together with families PF03547, PF1758 and PF13593 that is in agreement with our profile analysis. The connections between these families are found at e-value 1e-3.
(TIFF)

**S9 Fig. Projection of the phylogenetic tree of all 17 families studied here on the Tree of Life.** The phylogenetic tree (Fig 5B) is projected on the Tree of Life. Each protein family is colored by unique color similarly as in CLANS (Fig 5A). Presence of the protein family in a particular group of organism is shown by a cross sign "X".
(TIFF)

**S10 Fig. Comparison of unaligned full-length sequences in CLANS.** Full set of unaligned sequences of all 17 families were submitted into CLANS all-against-all BLAST search. A single sequence is represented by one "dot", circle. The sequences clustered in 3D based on pairwise similarity. Mostly, sequences clustered well into families similarly as was assigned by Pfam. Each Pfam family is shown in unique color. The same colors are used in profile-profile analysis (Fig 5A and S3 Fig). The largest (and also unknotted) family PF00999 was clustered into seven groups, all groups colored in blue. The sequences of slipknotted family PF03390 (colored in orange) clustered into one group. The families IDs with unknotted topologies are highlighted with blue font (PF00999, PF06965, PF01758). The family PF05145 from the clan CL0142 is colored red. The one-domain families IDs (LrgA—PF03788, LrgB—PF04172) are highlighted with magenta font. The fusion LrgA/LrgB that partially co-localize with PF04172 (LrgB) is shown in magenta. Some of one-domain LrgA (PF03788) sequences are located in between fusion protein and LrgA, marked with the star sign "*".
(TIFF)

**S11 Fig. Clustering-based characteristics matrix.** Characteristics matrix used for Bayesian phylogenetic trees generations. The matrix was generated based on the profile-profile connections at the lowest cut-off 1e-5 (Fig 5A and S3 Fig).
(TIFF)

**S1 File. Methods.** Sequence search. Sequence analysis. Procedure to identify the domains. Sequence profiles. Multiple sequence alignment. Phylogeny tree reconstruction. Visualization and figures preparation.
(PDF)

**S2 File. Multiple sequence alignment of domain sequences.**
(TXT)

**S3 File. Output from clustering of the domain sequences.**
(TXT)

**S4 File. Phylogenetic tree from MrBayes.**
(TXT)

## Author Contributions

**Conceptualization:** Stanislaw Dunin-Horkawicz, Joanna I. Sulkowska.

**Data curation:** Vasilina Zayats, Joanna I. Sulkowska.

**Formal analysis:** Vasilina Zayats, Aleksandra I. Jarmolinska.

**Funding acquisition:** Joanna I. Sulkowska.

**Investigation:** Vasilina Zayats, Aleksandra I. Jarmolinska, Borys Jastrzebski, Stanislaw Dunin-Horkawicz, Joanna I. Sulkowska.

**Methodology:** Vasilina Zayats, Stanislaw Dunin-Horkawicz.

**Software:** Aleksandra I. Jarmolinska, Stanislaw Dunin-Horkawicz.

**Supervision:** Stanislaw Dunin-Horkawicz, Joanna I. Sulkowska.

**Validation:** Aleksandra I. Jarmolinska, Stanislaw Dunin-Horkawicz, Joanna I. Sulkowska.

**Visualization:** Agata P. Perlinska, Borys Jastrzebski.

**Writing – original draft:** Vasilina Zayats.

**Writing – review & editing:** Vasilina Zayats, Agata P. Perlinska, Aleksandra I. Jarmolinska, Joanna I. Sulkowska.

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
