## [Decision Letter · Decision Letter 0]

15 Sep 2020

Dear DR Sulkowska,

Thank you very much for submitting your manuscript "Slipknotted and unknotted monovalent cation-proton antiporters evolved from a common ancestor" for consideration at PLOS Computational Biology.

As with all papers reviewed by the journal, your manuscript was reviewed by members of the editorial board and by three independent reviewers. In light of the reviews (below this email), we would like to invite the resubmission of a significantly-revised version that takes into account the reviewers' comments. 

Please note that all reviewers (but specially reviewer 1) pointed out deficiencies regarding the structure of your manuscript and presentation of the results that should be fully taken into account in the revised version. Note as well that there is missing information and data, which must be provided in the revised version, as well as the source code, which is not available.

We cannot make any decision about publication until we have seen the revised manuscript and your response to the reviewers' comments. Your revised manuscript is also likely to be sent to reviewers for further evaluation.

Sincerely,

Patrícia F. N. Faísca, Ph.D

Guest Editor

PLOS Computational Biology

Stefano Allesina

Deputy Editor

PLOS Computational Biology

Reviewer's Responses to Questions

**Comments to the Authors:**

Reviewer #1: In the manuscript “Slipknotted and unknotted monovalent cation-proton antiporters evolved from a common ancestor” by Zayats et al. the authors study the evolutionary origin of the slipknot topology in proteins. After identifying several protein families, which show (slip)knotted and unknotted topologies, their phylogenetic relationship and evolutionary history is analysed and evolutionary scenarios, leading to the slipknot topology coming from an unknotted ancestor, are inferred.

In its current form, the manuscript is unfortunately not suited to provide new insights into the origin(s) of this special slipknot topology, due to deficiencies in the presentation of the results.

The following comments about the, in my opinion, major issues and shortcomings of the manuscript hopefully help to clarify several points to make the manuscript finally comprehensible and the approach reproducible for a broader audience:

Language:

The manuscript needs excessive proof reading (preferably by a native speaker), since not only many spelling mistakes and wrong words or grammar hinder comprehension, but also lead to ambiguous meanings of sentences or render the understanding of presented results or conclusions impossible.

Additionally, many used words have specific meanings in the field of evolutionary biology (e.g. pathway), which differ from the context in which they are used here. This can lead to confusion and should be revised (see also Consistency and Structure paragraphs).

Consistency and definition of terms:

Many terms are not used consistently throughout the manuscript and make an understanding very difficult. Most of these terms are also never explained or just after they were used already multiple times (see also Structure paragraph). Especially the terms or IDs related to the Pfam database (Pfam, pfam, PFAM, PFam etc. all exist btw → consistency) are likely not known (with their definitions) by most people and should be explained properly to provide guidance to interpretation and comprehension of the findings. PFxxxx and CLxxxx are used for example extensively in sentences, but if I wouldn’t know that these are IDs of Pfam entries or corresponding Pfam clan IDs, respectively (which is never explained), the findings or conclusions based on this are not understandable at all. The same with the use of family/repeat/domain – I have the feeling here the HMM-types from the Pfam database are meant, but it is not common knowledge how these are defined within the Pfam database. Especially, the use of “repeat” should be carefully re-evaluated, since it is often not clear if a (tandem-)repeated domain in a certain HMM/family/protein is meant or a single instance of that “Repeat” (type in the Pfam database).

Colour-coding in and across figures could be improved to ensure consistency as well.

Structure:

Although most information is provided somewhere in the manuscript, it takes multiple readthroughs to find the according explanations or used methods and understand afterwards the related sentences.

If the Methods are at the very end of the manuscript, some simple explanations what has been done and how, needs to go into the Results section. The Results section contains furthermore at the beginning parts that are introduction and throughout the section parts, which are Discussion. Especially the presentation of the Results should be restructured completely (with consistency and definition of terms at first appearance in mind; see above).

Missing information:

Some information is missing completely and should be provided to make the conclusions comprehensible and reproducible. Especially in the beginning of the Results section: where have how many sequences been taken from and analysed/how big are the analysed families/what are their properties? Description and explanation of the methodology (how were the 7 evolutionary scenarios inferred?). The gitlab repo linked in the manuscript is not existent, therefore I couldn’t have a look at the source code and see what has been done. In many figures information or explanation is missing (double-check also Supplementary Info, S12 e.g. does not provide any information of what is shown (headers of the columns or description and meaning of the character states) etc. etc.). The role of repeats (and potential tandem repeats formed by tandem duplications rather than “fusion” of two single-domain proteins) should be explained and visualised. The provided Pfam-IDs can represent HMMs for example, which span already multiple instances of one repeat. How many repeat instances are included in the single-coloured bars in the figures or the according mentioned Pfam-IDs in the text does not become clear. Also, it should be clearly indicated in the text when conservation/similarity is discussed/considered from a structural or sequence perspective. Most conclusions made in the Results section are not comprehensible (probably because the methods behind it are missing) and should be elaborated. One of the conclusions is that observable patterns are biased “due to lack of data”; has any quality assessment been conducted to estimate this bias?

Finally, it should be made very clear what has been found for the first time in this study and what is actually the achievement of the authors in this study. The text reads in its current version very cryptic regarding the contribution (identification of a “new” slipknotted family, but all that based on existent Pfam-IDs and clans and PDB structures?), which makes it hard to distinguish between discussed former findings and the findings of this study.

Reviewer #2: In this manuscript, Zayats et al. investigated the evolution of a family of Slipknotted proteins, i.e., 2-hydroxy carboxylate transporter family. Importantly, they found distantly related protein families with unknotted topology and used this divergence to identify the scenarios behind the evolution of Slipknotted proteins. Overall, this is an interesting study and provides important details for the evolution of protein topology in a specific case. The manuscript, though, can benefit from a revision to enhance clarity and presentation. Here is the list of my major and minor comments:

Major comments:

1) The evolution of membrane transporters from one-domain proteins is not surprising and thoroughly discussed in the literature. I encourage authors to focus on the mechanisms of the divergence of Slipknotted and unknotted proteins and the fact that they could identify different scenarios that explain this divergence.

2) The major assumption behind this work is that the difference in the topology of slipknotted and unknotted proteins is caused by differences between TM helices and loops. Although this assumption sounds plausible based on previous studies (cited by the authors), authors should elaborate and justify why this assumption likely holds. They may use the current dataset and employ statistical analyses to see whether TMs and the loops contribute the most to such divergence.

Minor comments:

1) The manuscript is well-written but can greatly benefit from the substantiation of important concepts. For example, the authors should explain why knotted proteins are interesting and cover the most important literature on this problem.

2) Authors write "It was shown, that globular, as well as transmembrane, slipknotted proteins often appear in new organisms" in Lines 13-14. This is an overstated claim as the cited paper is only talking about the one protein family (2-Hydroxycarboxylate Transporter Family).

3) The domains and regions that were used for evolutionary analyses should be clarified in the text. The current writing can confuse the readers. For example, authors write in lines 48-49 "thus we used a single repeat from each structure for comparison". The sequence position should be stated clearly.

4) I found figures S2 and S3 (of the supplementary information) quite informative. Why don't authors bring up these figure to the main text and discuss them when they write about the superposition of knotted and unknotted proteins (Lines 48-63).

5) In constructing figures S5 and S6 and Fig. 3, it appears that the authors used the full sequence of N- and C-repeats. Would their clustering change if they only look at TMs and the connecting loops?

Reviewer #3: The article scrutinizes evolutionary relations between unknotted and slipknotted protein domains by using the sodium-citrate transporter as an example of a protein with a slipknotted domain. The topic of the article is very interesting, and it deserves publication in PLOC Computational Biology after revision.

Before the publication, several points should be addressed.

General Comments:

The manuscript should be amended to make it understandable to the general reader of PLOS Computational Biology.

Specifically, the difference between the unknotted, knotted and slipiknotted domains should be clearly explained in the Introduction section of the main text. The Fig. 1 is not quite successful. It does not explain how a knot is formed from a flat sequence profile shown on the top. Perhaps, intermediate steps of the knot formation should be shown. The Fig. S1 can be used in the main text to make the explanation clearer.

Also, the topology of the citrate transporter should be explained in more detail. This protein consists of functionally distinct scaffold and elevator domains. From the current text it is not clear whether all the analysis is related to the whole protein or only to its elevator domain. Therefore, the overall topology of the protein should be described and shown. Only after that, the evolutionary relation of the slipknotted (elevator?) domain to other proteins (domains?) should be analyzed in detail. Fig. S2 could be also used in the main text as an explanatory image albeit after modification. Currently there is hardly any difference between the panels S2A and S2C (left).

Generally, the authors should imagine a biologist without any idea about slipknotted domains and write for such a reader.

Specific Comments:

1. In the given family of transporters, the authors describe proteins with unknotted and slipknotted domains. It remains, however, unclear whether it is possible – topologically - to get to a slipknotted domain directly from an unknotted domain or the evolutionary transition should go via a knotted intermediate. Discussion of this point is highly desirable.

2. Line 104 and after: the complete multiple alignment with all amino acid sequences should be provided in Supplementary materials.

3. Fig. 3. Something is wrong with the color code in Fig. 3B. There are colors that are not mentioned in the figure caption.

4. Line 152: a word appears to miss after “widely”.

5. Line 167 and after: PDB IDs of the four known structures of relevant proteins should be explicitly given somewhere in the manuscript. The reference to Fig. 3B is not enough; neither the figure not the caption contains the PDB IDs.

6. The Discussion section is very brief. It might be interesting to discuss whether the slipknotting brings any functional advantages.

7. The discussion of PF03616, PF03601 and PF05684 should be moved from Conclusions into Discussion.

**Have all data underlying the figures and results presented in the manuscript been provided?**

Reviewer #1: **No: **The gitlab repo with the source code provided as a link in the manuscript does not exist.

Reviewer #2: Yes

Reviewer #3: **No: **The multiple amino acid sequence alignment is absent

PLOS authors have the option to publish the peer review history of their article (what does this mean?). If published, this will include your full peer review and any attached files.

Reviewer #1: No

Reviewer #2: No

Reviewer #3: No
---

## [Decision Letter · Decision Letter 1]

3 May 2021

Dear DR Sulkowska,

Thank you very much for submitting your manuscript "Slipknotted and unknotted monovalent cation-proton antiporters evolved from a common ancestor" for consideration at PLOS Computational Biology. As with all papers reviewed by the journal, your manuscript was reviewed by members of the editorial board and by several independent reviewers. The reviewers appreciated the attention to an important topic. Based on the reviews, we are likely to accept this manuscript for publication, providing that you modify the manuscript according to the review recommendations.

Sincerely,

Patrícia F. N. Faísca, Ph.D

Guest Editor

PLOS Computational Biology

Stefano Allesina

Deputy Editor

PLOS Computational Biology

[LINK]

Reviewer's Responses to Questions

**Comments to the Authors:**

Reviewer #2: I appreciate the authors' efforts to answer all the raised concerns. The manuscript is substantially improved and merits publication in PLoS Computational Biology. The figures' quality, however, could still be improved and I suggest authors and the editorial office to consider this issue.

Reviewer #3: General comments

1. The problem with the manuscript is still that some of its sections are incomprehensibly written. Apparently, experts in topology of knots, while applying their science to proteins, have developed their own terminology, which differs from that generally accepted in protein science. Since the manuscipt deals not only with the topology but also with the evolution of a large family of membrane transporters, the article should be understandable to researchers who work on these transporters. There is no reason to expect these researchers to know the mathematical theory of knots.

The terms "entlargement", "closed knot", "open knot" and "subchain of the backbone" are unusual in protein science and should be strictly defined in the text or replaced by more traditional and understandable analogues.

2. This reviewer recommends that the first paragraph of the results (or better, the whole text) be given to some biochemist to read and then edited according to the recommendations received.

3. The structures shown in Figures 3A and 3C are superposed in Fig. 3B. However, the green structure from Fig. 3A is painted blue in Fig. 3B, and the blue structure from Fig. 3C is painted green in Fig. 3B. The colouring of the structures in Fig. 3B should be changed.

Reviewer #4: This manuscript describes a broad family of transmembrane proteins where a slipknot has been found, along with homologues that are unknotted. Much of the paper deals with ideas about gene duplication and fusion for evolution in this family. The strength of the study is that the observation is intriguing, raising interesting evolutionary and functional questions. The limitation is that whether knotting features are important in proteins remains mainly speculative. Nonetheless, the evolutionary arguments are interesting, and the findings should be useful to those interested in transmembrane transporter function and evolution.

The revisions noted appear to have been largely effective at improving the original paper. Two issues for further improvement, one regarding substance and clarity and the other regarding presentation are:

1) The topological features that distinguish a slipknot from an unknot in a protein chain are hard to grasp even for experts and certainly for newcomers. The figures provided go part way to explaining things, but my sense is that most readers will remain somewhat befuddled. I suggest that the authors augment figure 4 with a simplified curve drawing for the knotted and unknotted curves, with a focus on explaining how two structurally similar cases can be topologically different. For example, can a diagram be drawn that might show that the difference arises from one structure having a curve crossing one way while the other structure has a curve crossing the other way. A diagram like this would go along way towards drawing in more readers with a connection between the sequence/structure evolution and the topological difference being reported.

2) The overall clarity of the paper is ok, and the mechanics are generally good. But some further editing is required. Maybe this can be done safely at the technical editor level. One point to focus on is the use of commas, which are not placed well in many instances (lines 188, 220, 308, and others). The negative use of “for none of these” in line 39 should be reworded. In line 143, ‘spacial’ should be ‘spatial’. In line 315, ‘constrains’ should be ‘constraints’. Words appear to be missing in line 327.

**Have the authors made all data and (if applicable) computational code underlying the findings in their manuscript fully available?**

Reviewer #2: Yes

Reviewer #3: Yes

Reviewer #4: Yes

PLOS authors have the option to publish the peer review history of their article (what does this mean?). If published, this will include your full peer review and any attached files.

Reviewer #2: No

Reviewer #3: No

Reviewer #4: No

Figure Files:

Data Requirements:

Reproducibility:

References:

---

## [Editor Report · Decision Letter 2]

28 Sep 2021

Dear DR Sulkowska,

We are pleased to inform you that your manuscript 'Slipknotted and unknotted monovalent cation-proton antiporters evolved from a common ancestor' has been provisionally accepted for publication in PLOS Computational Biology.

Best regards,

Patrícia F. N. Faísca, Ph.D

Guest Editor

PLOS Computational Biology

Stefano Allesina

Deputy Editor

PLOS Computational Biology

---

## [Editor Report · Acceptance letter]

11 Oct 2021

PCOMPBIOL-D-20-01125R2 

Slipknotted and unknotted monovalent cation-proton antiporters evolved from a common ancestor

Dear Dr Sulkowska,

I am pleased to inform you that your manuscript has been formally accepted for publication in PLOS Computational Biology. Your manuscript is now with our production department and you will be notified of the publication date in due course.

With kind regards,

Andrea Szabo
